# Germline Variants in the Immune Response-Related Genes: Possible Modifying Effect on Age-Dependent *BRCA1* Penetrance in Breast Cancer Patient

**DOI:** 10.3390/cancers17233756

**Published:** 2025-11-25

**Authors:** Ekaterina S. Kuligina, Aleksandr S. Martianov, Grigory A. Yanus, Yuliy A. Gorgul, Evgeny N. Suspitsin, Alexandr A. Romanko, Anastasia V. Tumakova, Alexandr V. Togo, Aniruddh Kashyap, Cezary Cybulski, Jan Lubiński, Evgeny N. Imyanitov

**Affiliations:** 1Laboratory of Molecular Oncology, Department of Tumor Growth Biology, N.N. Petrov National Medical Research Center of Oncology, Saint Petersburg 197758, Russia; 2Department of Medical Genetics, Saint Petersburg State Pediatric Medical University, Saint Petersburg 194100, Russia; 3International Hereditary Cancer Center, Pomeranian Medical University in Szczecin, 70-204 Szczecin, Poland

**Keywords:** breast cancer, *BRCA1*, germline mutations, PRF1, mutation penetrance, age at onset, inherited cancer risk, antitumor immunity

## Abstract

*BRCA1/2* mutations are associated with highly elevated but still not fatal risk of breast and ovarian cancer. The identification of modifiers *BRCA1/2* penetrance is essential for the personalization of medical management of carriers of *BRCA1/2* pathogenic alleles. There are two novelties related to this investigation. Firstly, all relevant studies in this area have a shortage of non-affected *BRCA1/2* mutation carriers. We have proposed an alternative approach, which is based on the comparison of early-onset versus late-onset cancer patients, thus assuming that the age at diagnosis may serve as a surrogate of *BRCA1/2* penetrance. Secondly, we focused not on common polymorphisms but on rare pathogenic variants in immune response-related genes, which cannot be analyzed upon conventional genome-wide association studies. This effort led to the demonstration of the *BRCA1* penetrance-modifying role for the *PRF1* p.Ala91Val variant. *PRF1* p.Ala91Val homozygotes are known to be affected by familial hemophagocytic lymphohistiocytosis, while heterozygous carriers of this allele may have a subclinical immune deficiency.

## 1. Introduction

*BRCA1/2* pathogenic variants are the most common genetic cause of hereditary breast cancer (BC) syndrome, being responsible for up to 5–8% of the total BC morbidity [1,2,3]. The penetrance of pathogenic *BRCA1/2* variants is usually within the range of 60–90%. Furthermore, *BRCA1/2* mutation carriers are characterized by huge variations with regard to the age of disease onset [4,5,6,7]. The individual lifetime risk of *BRCA1*-associated BC depends on the interplay between lifestyle and genetic factors, many of which are still unknown. There are several dozen single-nucleotide polymorphisms (SNPs), which have been identified mainly by the efforts of the Consortium of Investigators of Modifiers of *BRCA1* and *BRCA2* (CIMBA) as potential contributors to BC risk in *BRCA1/2* mutation carriers [8,9,10,11]. Many of them have been recommended for inclusion in “polygenic risk scores” to assist in the estimation of individual cancer susceptibility for *BRCA1/2*-carriers [12,13,14]. However, these SNP studies focused mainly on relatively common variations, e.g., SNPs with a minor allele frequency (MAF) exceeding 5%. Meanwhile, each individual genome contains a huge number of rare variants, many of which significantly affect the function of involved genes. These rare alleles cannot be efficiently analyzed by conventional genome-wide association studies (GWAS) and therefore require specifically designed investigations.

Immune-mediated mechanisms of tumor development may be particularly relevant to *BRCA1*-driven neoplasms, as *BRCA1* inactivation is associated with a deficiency in homologous DNA repair, chromosomal instability, and, consequently, increased tumor antigenicity [15,16,17]. There are several hundred genes whose homozygous inactivation is associated with primary immune deficiencies (PID), and which may, in theory, act as modifiers of *BRCA1/2* penetrance. We hypothesized that congenital defects in the functioning of some immune response genes may manifest as subclinical variations in immunity (including antitumor immunity) and influence the disease risk in *BRCA1/2* mutation carriers.

While designing the study, we aimed to overcome several limitations. First of all, the distribution of *BRCA1/2* pathogenic alleles is a subject of interethnic variations [18]. We acknowledged the low number of *BRCA2* mutation carriers in our collection as well as the differences between *BRCA1*- and *BRCA2*-driven cancers [19], and intentionally limited the study to the *BRCA1* gene. Secondly, while *BRCA1* mutations predispose both to breast and ovarian carcinomas, BCs are significantly more frequent [6,20]; therefore, we focused only on this category of cancer patients. Thirdly, the correct estimation of *BRCA1* penetrance requires a comparison of affected and non-affected subjects. The collection of a critical number of healthy persons with *BRCA1* mutations is complicated, especially given that “true” cancer-free status can be assigned only after achieving a certain age threshold. Furthermore, “control” *BRCA*1 mutation carriers are usually obtained via the analysis of family members of patients with *BRCA1*-driven cancer, which creates a bias. In our previous study, we have implemented an alternative approach for the analysis of *BRCA1* penetrance [21]. We assumed that *BRCA1* carriers with unfavorable genomic context are likely to develop cancer disease at an earlier age when compared to women with disease-protective allele combinations. Hence, the analysis of genotypes of young-onset vs. late-onset patients with breast cancer has a potential for identification of *BRCA1* penetrance modifiers. Here, we present the results of the study on pathogenic or likely pathogenic mutations in immune response genes in these patients’ groups.

## 2. Materials and Methods

We initially considered all patients with *BRCA1*-driven BC treated in the N.N. Petrov Institute of Oncology (St. Petersburg, Russia). The study was approved by the Local Ethics Committee (approval date: 22 October 2021; reference No. 36/302). To define the borderline values for the “young-onset” versus “late-onset” groups, we evaluated the age distribution of cancer manifestation in the hospital database, including the clinicopathological data on 1200 consecutive *BRCA1* mutation carriers who were forwarded to the local diagnostic facility between years 2008 and 2022 for genetic testing. The age cutoffs between the 1st and 4th quartiles were taken as thresholds, being <39 years for young onset and >57 years for late-onset. NGS for immune response genes was initially performed for 42 young-onset versus 35 late-onset *BRCA1*-mutated BC patients (the “discovery” cohort, Table 1; clinicopathological characteristics and the spectrum of *BRCA1* pathogenic variants for these patients are presented in Appendix A). The study workflow is shown in Figure 1.

High-molecular-weight blood-derived DNA served as a source for germline mutation analysis. The custom NGS panel included 353 genes associated with 354 inborn errors of immunity (IEIs), listed in the 2019 version of the IUIS classification [22]. The panel was designed via the NimbleDesign tool and ordered from Roche [23]. A list of genes and corresponding diseases is presented in Appendix A. DNA libraries were prepared using the Kapa HyperPlus Kit (Roche, Mannheim, Germany). Target enrichment was performed with the SeqCapEZ System (Roche, Mannheim, Germany). NGS analysis was carried out using the Illumina MiSeq platform with 70–90× coverage. The sequences were aligned to the GRCh37 (hg19) reference genome via the BWA 0.7.15 tool. Variant calling was performed using the GATK 3.6 instrument. Quality filtering was carried out with BCFtools 1.2 software. The multisample file was annotated using the snpEff v.4.3t tool [doi: 10.4161/fly.19695], and variants with predicted high or moderate impact were selected for further consideration. Clinical interpretation of the detected variants was performed under the ACMG/AMP 2022 guidelines. High priority was assigned to candidate mutations with the following pathogenicity attributes: (i) status “pathogenic/likely pathogenic” by the ClinVar database [24,25]; (ii) status “deleterious” by in silico tools CADD v1.7 (universal predictor; https://cadd.gs.washington.edu/, accessed on 18 November 2025) and fitCons V1.01 (the tool which integrates functional assays with selective pressure, http://compgen.cshl.edu/fitCons/, accessed on 18 November 2025); (iii) statistically increased mutation prevalence in the cancer cohort against the cancer-free population (MAF data presented in gnomAD, version 2); alleles producing OR per allele > 2 at *p* < 0.05 [26,27]; (iv) recurrent variants, which occurred two or more times in our collection, being exceptionally rare in the general population; and (v) potentially relevant functions of the candidate genes (i.e., involvement in the DNA damage response, proliferation, apoptosis, cell mobility, stress response, etc.).

Potentially relevant mutations were subjected to manual inspection using the Integrative Genomics Viewer (IGV) browser [https://bioviz.org/ (accessed on 15 June 2024)]. Wherever appropriate, variants were confirmed by Sanger sequencing.

Newly identified candidate variants were genotyped in the two-step validation study, which included 368 young-onset (<39 y.o.) and 427 late-onset (>57 y.o.) patients affected by *BRCA1*-driven BC (Figure 2). The group of “young” BC patients (median age: 33 years; range: 25–38 years) included 254 Russian women treated at the N.N. Petrov Institute of Oncology (SPb, St. Petersburg, Russia) and 114 Polish patients from Pomeranian Medical University (PUM, Szczecin, Poland). The “late-onset” group (median age: 61 years; range: 58–80 years) was composed of 326 Russian patients (“SPb”) and 101 Polish patients (“PUM”).

In the first step, the most promising candidate genetic variants selected upon NGS analysis were genotyped in 90 young-onset vs. 90 late-onset Russian BC patients using high-resolution melting (HRM) analysis coupled with Sanger sequencing of abnormally melted DNA fragments (“pilot” study). The oligonucleotides, which were utilized for this effort, are listed in Appendix A. *PRF1* p.Ala91Val genotype frequencies were subjected to second round of validation, which involved 278 young-onset and 337 late-onset Russian and Polish *BRCA1*-associated BC cases (the “enlarged” study).

The prevalence of candidate at-risk alleles and genotypes in the studied groups was statistically analyzed by the SPSS software (version 22) via two-sided Fisher’s exact, chi-square, or Mantel-Haenszel tests. The Bonferroni–Holm method was applied for multiple-test correction.

## 3. Results

NGS genotyping of 353 immune response genes was performed for 42 young-onset and 35 late-onset Slavic BC patients carrying *BRCA1* pathogenic alleles. A total of 2054 non-synonymous coding gene variants were detected (Appendix A). The process of variant filtering is described in Figure 3. We considered as candidates only rare (gnomAD MAF < 5%) protein-truncating variants (n = 80) and missense variants with in silico pathogenicity CADD scores >/= 25 (n = 105) (Appendix A). Further, we selected those mutations that were found exclusively in either young- or late-onset patients (Appendix A). According to our hypothesis, the “young-onset” associated alleles increase the penetrance of *BRCA1*, whereas the “late-onset” variants are likely to have a protective effect, delaying the age of BC manifestation.

Based on the predefined criteria, 29 variants were classified as likely “protective” alleles and 42 as potential modifiers increasing *BRCA1* penetrance (Appendix A). The prevalence of 22 top candidate variants was assessed in a “pilot” molecular epidemiological case–control study comprising 90 young-onset and 90 late-onset Russian *BRCA1*-driven BC cases (Table 2). Twelve of these variants were absent in both cohorts; apparently, their impact on *BRCA1*-driven BC risk could not be evaluated within a reasonably powered case-control study. Ten variants were observed at low frequencies (1–5%) and showed comparable distributions between the two groups, suggesting no apparent effect on the age of BC onset. Notably, only one variant, *PRF1* p.Ala91Val (rs35947132), was detected exclusively in the young-onset group, with a borderline statistically significant enrichment [7/73 (9.6%) vs. 0/78 (0%), Fisher’s exact test *p* = 0.005; *p* after adjustment for multiple comparisons = 0.055]. This mutation was selected for extended case-control analysis in independent *BRCA1*-mutated BC cases, which included 278 young-onset and 337 late-onset patients from Poland (“PUM” cohort) and Russia (“SPb” cohort) (Table 3).

In the “SPb” cohort, *PRF1* p.Ala91Val carriers occurred significantly more frequently among early-onset patients than among late-onset patients [14/164 (8.5%) vs. 8/236 (3.4%), *p* = 0.042, Fisher’s exact test]. The increase in the prevalence of the *PRF1* p.Ala91Val allele in the early-onset group was also statistically significant [16/328 (4.9%) vs. 8/472 (1.7%), *p* = 0.01, Fisher’s exact test]. This trend was not replicated in the “PUM” cohort, although a borderline numerical increase in the frequency of the *PRF1* p.Ala91Val allele and corresponding genotypes was also observed. The significance of the association between the presence of the *PRF1* p.Ala91Val allele and earlier age of BC manifestation [24/278 (8.6%) vs. 15/337 (4.4%), *p* = 0.045] was retained upon the pooled analysis of both groups (Table 3). The adjusted OR per carrier was 1.9 [1.00–3.76], and the OR per allele was 2.14 [1.13–4.07] (*p* = 0.068 and *p* = 0.024, Mantel Haenszel chi-square test).

For comparison, we analyzed a cohort of *BRCA1*-positive breast cancer patients with ages at onset ranging from 39 to 57 years (n = 84). In this intermediate group, the prevalence of heterozygous *PRF1* p.Ala91Val genotypes was 5/84 (6.0%), with no homozygotes detected. As expected, this frequency falls between those observed for the young-onset (8.6%) and late-onset (4.4%) patient groups.

## 4. Discussion

*PRF1* gene encodes perforin, which is a toxin responsible for the lysis of infected or neoplastic cells. Perforin induces the formation of pores in the attacked cell and acts in combination with protease granzymes. These proteins are stored in specialized secretory lysosomes, which are characteristic of cytotoxic T lymphocytes and “natural killers” (NKs) [28]. When these lytic granules are released, targeted cells undergo apoptosis [29,30]. Biallelic germline inactivation of the *PRF1* gene (10q22.1) is associated with familial hemophagocytic lymphohistiocytosis (FHL), a severe hereditary syndrome characterized by excessive inflammation; this condition is caused by the inability of NK and CD8+ T cells to eliminate target cells through perforin-dependent and granule-mediated cytotoxicity [31,32]. Partial perforin deficiency, which is likely to be observed in heterozygous carriers of *PRF1* alleles with impaired function, may be the cause of delayed FHL or other inflammatory or neoplastic disorders [33].

Our study demonstrated that the prevalence of the p.Ala91Val missense variant in the *PRF*1 gene was nearly twofold greater among young-onset *BRCA1*-driven BC patients than among late-onset patients (4.9% vs. 2.2%). The hypomorphic p.Ala91Val allele (rs35947132) is the most common variant observed in the Caucasian population, with a minor allele frequency (MAF) of 7–9%. The pathogenic potential of this mutation is well documented. The monoallelic p.Ala91Val substitution results in a 10-fold reduction in the lytic activity of the enzyme due to protein misfolding, leading to a notable decline in NK-cell cytotoxicity in healthy carriers of this variant [34,35]. Individuals who are homozygous for p.Ala91Val develop familial hemophagocytic lymphohistiocytosis type 2 (FHL2), although the penetrance of this genotype is not complete [36]. Some data indicate that *PRF1* p.Ala91Val heterozygosity is associated with subclinical immunodeficiency symptoms [37]. Notably, *PRF1* p.Ala91Val has also been linked to an increased risk of various hematological cancers, including B and T-cell lymphoma and acute lymphoblastic leukemia [38,39,40]. While the presence of a single copy of the p.Ala91Val allele is unlikely to significantly impact cancer risk, it may compromise the antitumor immune response, potentially contributing to the earlier development of certain solid tumors that are known to be controlled by the immune system, such as *BRCA1*-driven breast carcinomas. One may hypothesize that in young females, where hormonal stimulation drives rapid mammary epithelial proliferation, the combined effects of defective DNA repair resulting from *BRCA1* dysfunction and compromised immune surveillance due to *PRF1* deficiency may synergistically accelerate oncogenic transformation. Furthermore, subclinical impairment of natural killer cell function and other cytotoxic effector mechanisms associated with *PRF1* p.Ala91Val heterozygosity may promote the release of pro-inflammatory cytokines, including TNF, IL-6, and IFN-γ [41,42], fostering a microenvironment conducive to tumor initiation and progression. The elevated incidence of hematologic malignancies among *PRF1* p.Ala91Val carriers further supports the broader oncogenic potential of partial perforin deficiency [38,39,40].

In the discovery cohort, three carriers of the *PRF1* p.Ala91Val variant were identified within the young-onset group; two of these cases (67%) exhibited the Luminal A subtype, compared with 14 of 51 (27%) among *PRF1* wild-type cases (Table 1). Among the five additional *PRF1* carriers with available receptor status information, four presented with Luminal A tumors. These trends are interesting, given that usually no more than quarter of *BRCA1*-associated tumors belong to the Luminal A subtype [43,44,45].

This study has several limitations. It is noteworthy that although the potential significance of the *PRF1* p.Ala91Val allele has been proven by statistical analysis, its role was highly evident only in Russian patients but not in Polish patients. These discrepancies may result from random variation or reflect the influence of local environmental factors and underlying ethnic heterogeneity. The Russian cohort is likely to involve individuals with diverse ancestral backgrounds, including populations from the Caucasus, Siberia, and the Far East, whereas the Polish cohort is apparently more genetically homogeneous, consisting predominantly of women of Western Slavic descent [46]. The role of geographic, ethnic or other variations in determining *BRCA1/2* penetrance has not yet been properly addressed in available studies [47,48,49,50]. Furthermore, although the distribution of the *PRF1* p.Ala91Val alleles appears to differ between early-onset and late-onset *BRCA1*-associated BC patients, these data do not strictly support the penetrance-modifying role of this genetic variation. Finally, as this study focused exclusively on genes associated with inborn errors of immunity, it cannot exclude the possibility that additional generic modifiers may influence the age-related BC risk in *BRCA1* mutation carriers [51]. Properly designed case-control multicenter investigations are needed for the validation of the role of variations in immune-related genes in determining *BRCA1* penetrance.

## 5. Conclusions

Our findings suggest that rare genetic variations may play a role in modulating cancer risk among *BRCA1/2* mutation carriers, underscoring the potential value of including these variants in future analyses. Previous investigations revealed that pathogenic variants in low-penetrance cancer-predisposing genes contribute to breast cancer development in individuals with *BRCA1/2* mutations [52,53,54,55,56]. In this study, we obtained suggestive evidence for the penetrance-modifying role of genes associated with primary immune deficiencies. Further systematic exome-wide comparisons between affected and unaffected *BRCA1/2* heterozygotes, or between young-onset and late-onset *BRCA1/2*-driven cancer cases, are required to reveal novel genetic determinants which influence cancer risk *BRCA1/2* mutation carriers.

## Figures and Tables

**Figure 1 cancers-17-03756-f001:**
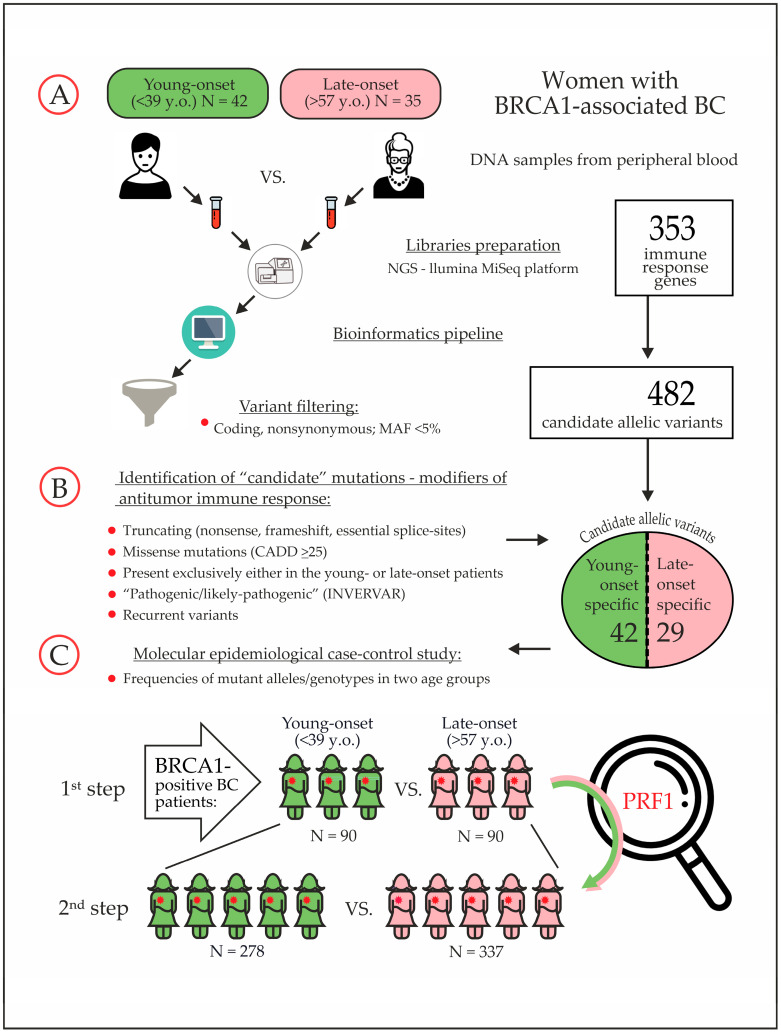
The workflow of the study: NGS screening of immune response genes and molecular epidemiological evaluation of selected candidate modifiers of *BRCA1*-driven BC risk.

**Figure 2 cancers-17-03756-f002:**
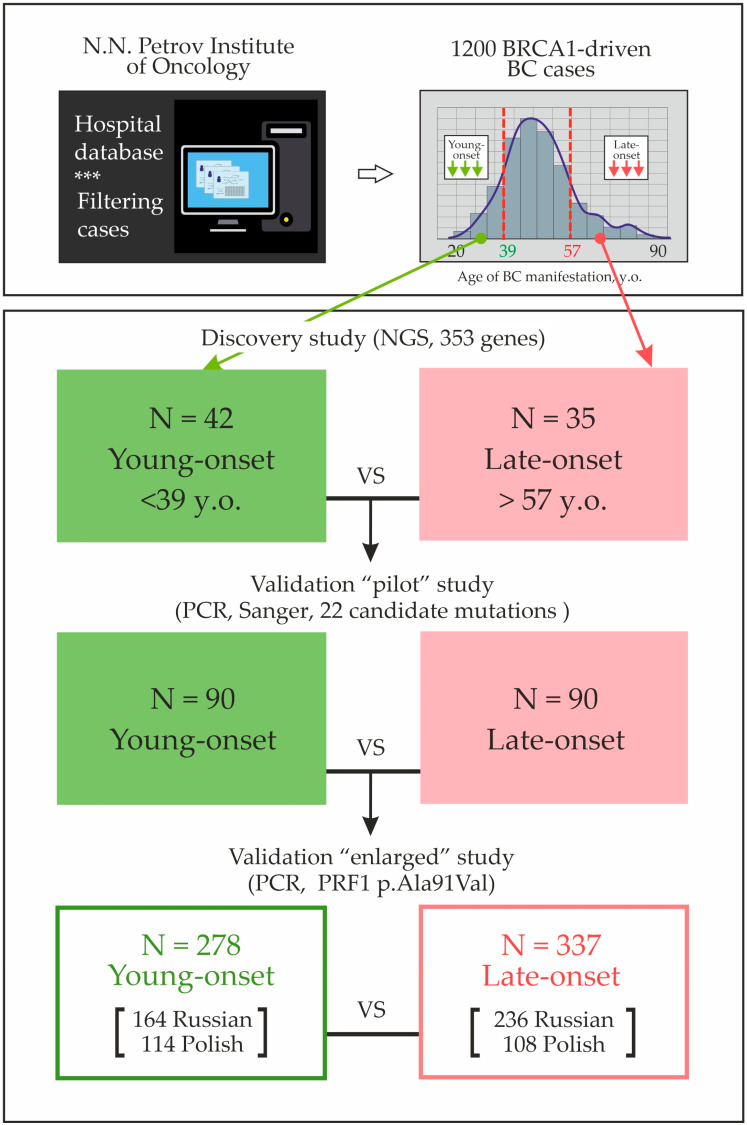
BC patients enrolled in the discovery and validation studies.

**Figure 3 cancers-17-03756-f003:**
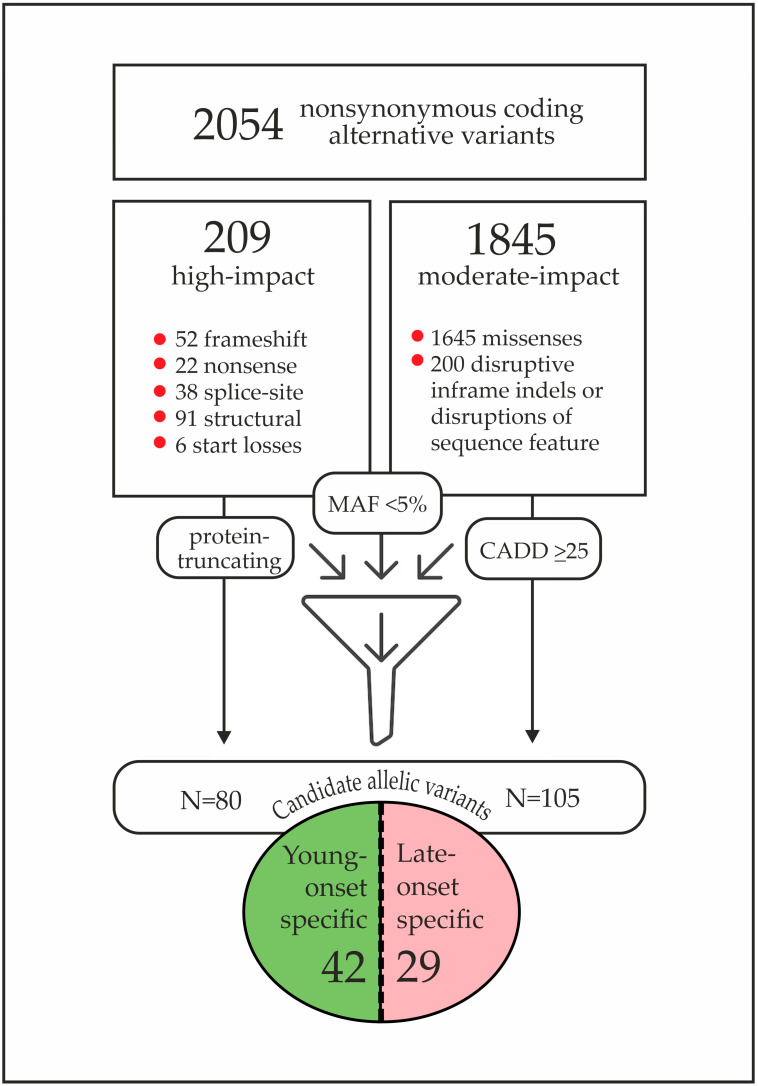
Results of the targeted sequencing of 353 immune-response genes: selection of candidate variants for the extended analysis.

**Table 1 cancers-17-03756-t001:** Characteristics of patients with *BRCA1*-driven breast cancer (BC) subjected to targeted NGS of immunodeficiency genes (the “discovery” cohort).

#	Age Group	Age y.o.	*BRCA1* Mutation	Family History	BC Sub Type	Candidate Variant from the “Discovery” Study
1	late-onset	58	*BRCA1* p.Ala457fs	s-BC 49, m-ThC 49	TNBC	
2	late-onset	66	*BRCA1* 5382insC	m-OC 50	nd	
3	late-onset	76	*BRCA1* 5382insC	d-BC 49	LumA	
4	late-onset	61	*BRCA1* 5382insC	m-BC, d-BC	nd	*DOCK8* p.Tyr1340Cys
5	late-onset	58	*BRCA1* 5382insC	m-BC, aunt(f)	TNBC	
6	late-onset	83	*BRCA1* 5382insC	d-BC(BRCA), s-GaCa	LumA	
7	late-onset	64	*BRCA1* 5382insC	No	nd	*DDX58* rs61752945; *DDX41* p.Val408Asp
8	late-onset	62	*BRCA1* 5382insC	No	nd	
9	late-onset	70	*BRCA1* 5382insC	f-CRC, m-BC, aunt(f)-BC	LumA	*TPP1* p.Arg208Ter
10	late-onset	60	*BRCA1* 5382insC	No	TNBC	
11	late-onset	60	*BRCA1* 5382insC	m-OC, s-OC	TNBC	
12	late-onset	68	*BRCA1* 5382insC	No	LumA	*AK2* rs138577419; *DNAJC21* p.Arg539Gln
13	late-onset	61	*BRCA1* 5382insC	aunt(f)-BC	nd	*SP110* p.Gly483Arg
14	late-onset	64	*BRCA1* 5382insC	s-BC 50	TNBC	
15	late-onset	65	*BRCA1* 5382insC	f-LC	nd	*JAGN1* p.Met1?
16	late-onset	61	*BRCA1* 5382insC	m-BC, aunt(m)-UtCa	TNBC	
17	late-onset	62	*BRCA1* 5382insC	No	TNBC	
18	late-onset	61	*BRCA1* 5382insC	f-EsophCa, aunt (m)-BC 56	LumA	
19	late-onset	61	*BRCA1* 4153delA	f-HNSSC	nd	
20	late-onset	68	*BRCA1* 5382insC	m-GaCa, aunt(f), uncle(m)-GaCa	TNBC	*DDX41* p.Val408Asp
21	late-onset	62	*BRCA1* 5382insC	m-BC, f-CRC, gm(m)-OC	TNBC	
22	late-onset	58	*BRCA1* 5382insC	s-BC	LumB	*DDX41* p.Val408Asp
23	late-onset	64	*BRCA1* 4153delA	gm(m)-GaCa 60, uncle(m)-GaCa 60, aunt(f)-UtCa 60	TNBC	*TMC6* p.Pro502Leu
24	late-onset	63	*BRCA1* p.L1205fs	m-BC 38	nd	
25	late-onset	61	*BRCA1* 4153delA	m-UtCa	LumA	*IL12B* rs3213119
26	late-onset	59	*BRCA1* 5382insC	no	LumA	
27	late-onset	61	*BRCA1* p.D435fs	m-BC 53, b-BraCa 29	LumA	
28	late-onset	61	*BRCA1* 5382insC	no	nd	*DDX58* rs61752945
29	late-onset	59	*BRCA1* 4153delA	no	nd	
30	late-onset	60	*BRCA1* 5382insC	aunt(m)-GaCa	nd	
31	late-onset	61	*BRCA1* 5382insC	nd	nd	*IL12B* rs3213119; *TMC6* p.Pro502Leu
32	late-onset	61	*BRCA1* 5382insC	nd	nd	
33	late-onset	59	*BRCA1* 5382insC	no	nd	
34	late-onset	63	*BRCA1* 5382insC	no	nd	
35	late-onset	62	*BRCA1* 5382insC	m-BC	TNBC	*ATP6AP1* p.Arg15Ter
36	young-onset	27	*BRCA1* 5382insC	gm(m)-LC 45	LumA	*NOP10* p.Asp12His; *NLRP1* p.Phe629Leu
37	young-onset	36	*BRCA1* 2080delA	m-BC 62, gm(m)-BC 70	TNBC	
38	young-onset	37	*BRCA1* 185delAG	m-BC, s-BC	nd	*PEPD* p.Arg237Cys
39	young-onset	35	*BRCA1* c.3629_3630delAG	nd	TNBC	
40	young-onset	38	*BRCA1* 5382insC	m-BC 55, gm(m)-OC 48, aunt(f)-OC 50, gf(f)-HCC 80, s-ThC 32	LumA	
41	young-onset	38	*BRCA1* c.5215+1G>T	s-BC 40	TNBC	
42	young-onset	33	*BRCA1* c.3304_3307delAATT	m-BC	TNBC	*NOP10* p.Asp12His
43	young-onset	36	*BRCA1* 4153delA	m-BiBC, OC 30, aunt(m)-CRC, gf(m)-LC 70	nd	
44	young-onset	27	*BRCA1* p.S281fs	m-OC 36, gm(f)-RenC	TNBC	*STAT4* p.Thr446Ile
45	young-onset	30	*BRCA1* 5382insC	no	LumB	*NOP10* p.Asp12His
46	young-onset	27	*BRCA1* 5382insC	m-BC 46, gm(m)-CRC 55, gm(f)-HCC 60	HER2+++	
47	young-onset	29	*BRCA1* p.R1726fs	no	TNBC	
48	young-onset	29	*BRCA1* C61G	m-BC	TNBC	
49	young-onset	30	*BRCA1* 5382insC	m-BC, gm(m)-BC	TNBC	
50	young-onset	30	*BRCA1* 5382insC	aunt(m)-CaUt	nd	*RORC* p.Arg10Ter; *NLRC4* p.Arg310Ter
51	young-onset	26	*BRCA1* 5382insC	aunt(f)-BC, gf(m)-LC	TNBC	
52	young-onset	27	*BRCA1* 5382insC	gm(f)-small intestine Ca	TNBC	*PRF1* p.Ala91Val
53	young-onset	34	*BRCA1* 5382insC	gm-BC	nd	
54	young-onset	28	*BRCA1* 5382insC	gm(f)-BC, aunts (m,f)-BC	TNBC	
55	young-onset	33	*BRCA1* 5382insC	no	LumB	*NOD2* p.Gly908Arg
56	young-onset	31	*BRCA1* 5382insC	m-BC 27, gm(m)-BC	TNBC	*NOD2* p.Gly908Arg
57	young-onset	37	*BRCA1* 5382insC	no	LumA	
58	young-onset	35	*BRCA1* 5382insC	gm(m)-melanoma 60, gm(f)-BC 50	TNBC	
59	young-onset	36	*BRCA1* 4153delA	aunt(f)-OC 45, gm(f)-CRC 68, gf(f)-CaLarynx 62, aunt(cousin,f)-OC 55	LumA	*PRF1* p.Ala91Val (homo)
60	young-onset	37	*BRCA1* 5382insC	aunt(f)-GaCa 39	LumA	
61	young-onset	36	*BRCA1* 5382insC	m-BC 38, aunt(m)-BraCa 42	nd	
62	young-onset	27	*BRCA1* 5382insC	no	TNBC	
63	young-onset	34	*BRCA1* c.5075-1G>C	m-LC 54, aunt(m)-BC 60	LumA	
64	young-onset	31	*BRCA1* 5382insC	gf(m)-LC	TNBC	
65	young-onset	37	*BRCA1* 5382insC	gm(m)-BC 43	nd	*PNP* rs104894453
66	young-onset	31	*BRCA1* 4153delA	m-BC, s-BC, gm(m)-OC	nd	*IL17RC* rs148575246
67	young-onset	32	*BRCA1* 5382insC	m-OC	TNBC	
68	young-onset	34	*BRCA1* 5382insC	f-PrC	nd	
69	young-onset	34	*BRCA1* 5656del20	no	TNBC	
70	young-onset	33	*BRCA1* 5382insC	gf(f)-GaCa	LumA	
71	young-onset	38	*BRCA1* 5382insC	m-BC, s-BC, aunt-BC	TNBC	
72	young-onset	32	*BRCA1* 5382insC	f-CaLarynx, ggm(m)-BraCa	TNBC	*RORC* p.Arg10Ter
73	young-onset	35	*BRCA1* 5382insC	gm(f)-BC	TNBC	
74	young-onset	33	*BRCA1* 5382insC	gm-CRC, gf(m)-LC	LumA	*PRF1* p.Ala91Val
75	young-onset	37	*BRCA1* 5382insC	gm(f)-PanCa	TNBC	
76	young-onset	32	*BRCA1* 5382insC	nd	TNBC	
77	young-onset	38	*BRCA1* 5382insC	m-BC	TNBC	

BC—breast cancer, ThC—thyroid cancer, OC—ovarian cancer, GaCa—gastric cancer, CRC—colorectal cancer, UtCa—uterine cancer; EsophCa—esophageal cancer, HNSSC—head and heck squamous cell carcinoma, BraCa—brain cancer, HCC—hepatocellular carcinoma, LC—lung cancer, RenC—renal cancer, PrC—prostate cancer, PanCa—pancreatic cancer, TNBC—triple-negative BC, LumA—luminal A BC, LumB—luminal B BC, m—mother, gm—grandmother, f—father, gf—grandfather, s—sister, b—brother, d—daughter.

**Table 2 cancers-17-03756-t002:** The frequency of 22 top candidate allelic variants discovered by NGS in 90 young-onset vs. 90 late-onset *BRCA1*-driven BC patients (the “pilot” study).

Gene	Description	Protein/ rs-id dbSNP	Effect	CADD *	fitCons **	MAF, %	Discovery Study N Carriers		Validation “Pilot” Study [wt/wt-mut/wt-mut/mut]	
							**Young-** **Onset** **(<39 y.o.)**	**Late-** **Onset** **(>57 y.o.)**	**Young-** **Onset** **(<39 y.o.)**	**Late-** **Onset** **(>57 y.o.)**
*AK2*	Adenylate kinase 2	-/ rs138577419	Structural interaction	25.8	0.73 (del)	0.221	0	1	90-0-0	90-0-0
*SP110*	SP110 nuclear body protein	p.Gly483Arg/rs149485401	Missense	28.9	0.72 (del)	0.949	0	1	90-0-0	90-0-0
*JAGN1*	Jagunal homolog 1	p.Met1?/rs143438463	Start lost	25.9	0.44 (be)	0.132	0	1	84-0-0	79-0-0
*IL12B*	Interleukin 12B	-/ rs3213119	Structural interaction	25	0.53 (del)	3.002	0	2	88-2-0	98-0-0
*DOCK8*	Dedicator of cytokinesis 8	p.Tyr1340Cy/rs116920018	Missense	32	0.71 (del)	0.327	0	1	89-1-0	98-0-0
*DDX58*	DExD/H-box helicase 58	-/ rs61752945	Structural interaction	27.7	0.71 (del)	1.927	0	2	83-1-0	86-2-0
*TPP1*	Tripeptidyl peptidase 1	p.Arg208Ter/rs119455955	Stop gained	36	0.72 (del)	0.04	0	1	88-2-0	90-0-0
*TMC6*	Transmembrane channel like 6	p.Pro502Leu/rs75400929	Missense	25.8	0.71 (del)	0.976	0	2	89-0-0	90-0-0
*ATP6AP1*	ATPase H+ transporting accessory protein 1	p.Arg15Ter/rs201620814	Stop gained	28.8	n/a	0.342	0	1	90-0-0	90-0-0
*DDX41*	DEAD-box helicase 41	p.Val408Asp/no ID	Missense	32	0.71 (del)	.	0	3	87-3-0	87-3-0
*DNAJC21*	DnaJ heat shock protein family (Hsp40) member C21	p.Arg539Gln/rs146933471	Missense	33	0.74 (del)	0.053	0	1	90-0-0	90-0-0
*RORC*	RAR related orphan receptor C	p.Arg10Ter/rs17582155	Stop gained	36	0.5 (be)	0.368	2	0	90-0-0	90-0-0
*NLRC4*	NLR family CARD domain containing 4	p.Arg310Ter/rs199475953	Stop gained	35	0.55 (del)	0.031	1	0	90-0-0	90-0-0
*STAT4*	Signal transducer and activator of transcription 4	p.Thr446Ile/rs141331848	Missense	34	0.62 (del)	0.11	1	0	86-0-0	88-0-0
*IL17RC*	Interleukin 17 receptor C	-/ rs148575246	Splice donor	29.7	0.11 (be)	1.02	1	0	89-0-0	89-1-0
*PNP*	Purine nucleoside phosphorylase	-/ rs104894453	Structural interaction	26.5	0.67 (del)	0.004	1	0	89-0-0	86-0-0
*NOP10*	NOP10 ribonucleoprotein	p.Asp12His/rs146261631	Missense	28	0.44 (be)	1.218	3	0	87-3-0	87-3-0
*NOD2*	Nucleotide binding oligomerization domain containing 2	p.Gly908Arg/rs2066845	Missense	29.8	0.56 (del)	1.427	2	0	89-1-0	82-5-0
*NLRP1*	NLR family pyrin domain containing 1	p.Phe629Leu/rs149035689	Missense	25.9	0.71 (del)	1.225	1	0	90-0-0	90-0-0
*PEPD*	Peptidase D	p.Arg237Cys/rs766107449	Missense	33	0.71 (del)	0.002	1	0	89-0-0	90-0-0
*PRF1*	Perforin 1	p.Ala91Val/rs35947132	Missense	25	0.55 (del)	4.662	3	0	67-7-0 ***	78-0-0
*AIRE*	Autoimmune regulator	p.Arg257Ter/rs121434254	Stop gained	39	n/a	0.061	1	0	88-2-0	89-1-0

* CADD—universal tool for scoring the deleteriousness of single nucleotide variants, multi-nucleotide substitutions, and insertion/deletions variants in the human genome [https://cadd.gs.washington.edu/ (accessed on 10 May 2025)]; ** fitCons—in silico pathogenicity predictor, which integrates the results of functional assays (such as ChIP-Seq) with the data on selective pressure [http://compgen.cshl.edu/fitCons/ (accessed on 23 October 2025).], del—deleterious (score > 0.50), be—benign (score ≤ 0.50); *** Statistically significant differences between age groups have been detected: 7/73 (9.6%) vs. 0/78 (0%), Fisher exact test *p* = 0.005.

**Table 3 cancers-17-03756-t003:** The frequency of *PRF1* p.Ala91Val mutations in *BRCA1*-associated BC patients with early- vs. late disease onset (the “enlarged” study).

Groups	Age at Onset	*PRF1* p.Ala91Val Carriers (%)	Total (Patients)	*p*- Value *	OR (95% CI) per Carrier	*PRF1* p.Ala91Val mut/mut (%)	Total (Patients)	*p*-Value *	OR (95% CI) per Homozygote	*PRF1* p.Ala91Val alleles (%)	Total (Alleles)	*p*- Value *	OR (95% CI) per Allele
BC SPb	<39 y.o. young- onset	14 (8.5%)	164	0.042	2.7 [1.08–6.51] *p* = 0.032	2 *** (1.2%)	164	0.047	7.1 [0.35–152.58] *p* = 0.201	16 (4.9%)	328	0.01	3.0 [1.25–7.03] *p* = 0.013
	>57 y.o. late- onset	8 (3.4%)	236			0 (0%)	236			8 (1.7%)	427		
BC PUM	<39 y.o. young- onset	10 (8.8%)	114	0.800	1.3 [0.47–3.52] *p* = 0.618	1 *** (0.9%)	114	1.000	2.7 [0.11–66.60] *p* = 0.540	11 (4.8%)	228	0.630	1.4 [0.54–3.71] *p* = 0.484
	>57 y.o. late- onset	7 (7.6%)	101			0 (0%)	101			7 (3.5%)	202		
BC PUM + SPb	<39 y.o. young- onset	24 (8.6%)	278	0.045	1.9 ** [1.00–3.76] *p* = 0.068	3 *** (1.1%)	278	0.273	8.6 [0.44–146.55] *p* = 0.156	27 (4.9%)	556	0.017	2.14 ** [1.13–4.07] *p* = 0.024
	>57 y.o. late- onset	15 (4.4%)	337			0 (0%)	337			15 (2.2%)	674		

* Fisher exact test; ** OR adjusted, Mantel Haenszel chi-square test; *** The carriers of homozygous *PRF1* p.Ala91Val genotype manifested with BC at age 27, 34, and 36, respectively.

## Data Availability

The original contributions presented in the study are included in the article/Appendix A; further inquiries can be directed to the corresponding author.

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
