# Peer review of "Germline Variants in the Immune Response-Related Genes: Possible Modifying Effect on Age-Dependent *BRCA1* Penetrance in Breast Cancer Patient"

_cancers, 2025, doi:10.3390/cancers17233756_

Round 1

Reviewer 1 Report

Comments and Suggestions for Authors

In the article ‘Germline variants in the immune response-related genes: possible modifying effect on age-dependent BRCA1 penetrance in 3 breast cancer patients’ Ekaterina et al performed a case-control comparison of affected BRCA vs non-affected BRCA mutation in breast cancer between young onset (<39 y.o.) and late onset (>57 y.o.). The authors identified 22 potentially relevant variants. Of note is the identification of PRF1p.Ala91Val variant.

The inclusion of a paragraph on limitations in the study in the introduction helps to understand potential biases before proceeding.

Major Concerns

The study did not consider any potentially co-occurring mutations with BRCA1in the study group hence its challenging to determine if the variants identified can only be truly linked to BRCA1 mutation.

1. Maybe helpful to generate a summary of supplementary table 1 on clinicopathological data that is included in the main text.

2. It maybe helpful for the authors to determine if the occurrence of the PRF1 pAla91val variant is linked to molecular subtypes of BC.

3. The authors distinguished and present data related to young- and late- onset, however a large cohort of patients within the ages 39-57 were not covered, hence would be important if analysis on the presence of the PRF1 variant can be established in this group.

4. The absence of PFR1 variant significance in ‘PUM’ cohort raises questions if the presence of PFR1 variant can be ethical based.

 Minor

5. Supplementary table S4 is not non-synonymous mutations but rather primer & probes design.

6. Important to include a list of 80 protein-truncating variants and 105 missense variants should be provided. Particularly a list of variants found exclusively in young-onset vs late-onset.

7. Line 16/17 in results seems more like classifications methods and should be properly described in methods.

8. Supplementary table S5 does show differentially expressed genes in young vs early onset, description of ‘protective’ and ‘potential modifiers’ can not be seen in the table.

Author Response

Comment: The study did not consider any potentially co-occurring mutations with BRCA1in the study group hence its challenging to determine if the variants identified can only be truly linked to BRCA1 mutation.

Response: This limitation is relevant to all case-control studies on genetic modifiers of BRCA-associated cancer risks. 

Comment: Maybe helpful to generate a summary of supplementary table 1 on clinicopathological data that is included in the main text.

Response: We have incorporate a relevant table in the Materials and Methods.

Comment: It maybe helpful for the authors to determine if the occurrence of the PRF1 pAla91val variant is linked to molecular subtypes of BC.

Response: We now comment on this in the Discussion (see below). To our opinion, the amount of relevant data is not sufficient to present them in the Results

“In the discovery cohort, three carriers of the PRF1 p.Ala91Val variant were identified within the young-onset group; two of these cases (67%) exhibited the Luminal A subtype, compared with 14 of 51 (27%) among PRF1 wild-type cases (Table 1). Among the five additional PRF1 carriers with available receptor status information, four presented with Luminal A tumors (data not shown). These trends are interesting, given that usually no more than quarter of BRCA1-associated tumors belong to the Luminal A subtype [43-45].”

Comment: The authors distinguished and present data related to young- and late- onset, however a large cohort of patients within the ages 39-57 were not covered, hence would be important if analysis on the presence of the PRF1 variant can be established in this group.

Response: We have analyzed some additional samples and now present these data in the Results:

“For comparison, we analyzed a cohort of BRCA1-positive breast cancer patients with ages at onset ranging from 39 to 57 years (n = 84). In this intermediate group, the prevalence of heterozygous PRF1 p.Ala91Val genotypes was 5/84 (6.0%), with no homozygotes detected. As expected, this frequency falls between those observed for the young-onset (8.6%) and late-onset (4.4%) patient groups.”

Comment: The absence of PFR1 variant significance in ‘PUM’ cohort raises questions if the presence of PFR1 variant can be ethical based.

Response: We now comment on this in the Discussion:

“…These discrepancies may result from random variation or reflect the influence of local environmental factors and underlying ethnic heterogeneity. The Russian cohort is likely to involve individuals with diverse ancestral backgrounds, including populations from the Caucasus, Siberia, and the Far East, whereas the Polish cohort is apparently more genetically homogeneous, consisting predominantly of women of Western Slavic descent [46]….”

Comment: Supplementary table S4 is not non-synonymous mutations but rather primer & probes design.

Response: Thank you, we have corrected this error.

Comment: Important to include a list of 80 protein-truncating variants and 105 missense variants should be provided. Particularly a list of variants found exclusively in young-onset vs late-onset.

Response: We have a created an addition Supplementary Table S3b, which contains this information.

Comment: Line 16/17 in results seems more like classifications methods and should be properly described in methods.

Response: This paragraph has been moved to the “Materials and Methods”

Comment: Supplementary table S5 does show differentially expressed genes in young vs early onset, description of ‘protective’ and ‘potential modifiers’ can not be seen in the table.

Response: A description has been added below the table, and an additional column titled “Expected effect on BRCA1-associated BC risk” has been incorporated.

Reviewer 2 Report

Comments and Suggestions for Authors

In "Germline variants in the immune response-related genes: possible modifying effect on age-dependent BRCA1 penetrance in breast cancer patients", Kuligina and colleagues present evidence for an involvement of germline gene variants in BRCA1 penetrance in breast cancer. They use DNA sequence analysis of  (relatively small) BRCA1-mutant breast cancer samples to find gene mutations with a significantly enhanced presence when compared to normal tissue samples. 

This is a very interesting premise, but the manuscript is not yet suitable for publication because of these issues:

  1. The discovery and validation cohorts are (too) small.
  2. No comparison is made with larger cohorts in the public domain (e.g. TCGA, cBioportal).
  3. There is no extensive bio-informatic or (even preliminary) functional analysis on the gene variants found. No discussion is provided for the variants from the literature (also from other cancers?).
  4. This makes it difficult to re-construct how these variants could influence the tumorigenesis.
  5. The Materials and Methods section is way too short & superficial.
  6. The figures, though descriptive, are large and not consistent in style, font, etc. This makes comparison difficult.

Author Response

Comment: The discovery and validation cohorts are (too) small.

Response: We have enlarged the study by adding 235 (114 young-onset and 119 late-onset) cases to the data set.

Comment: No comparison is made with larger cohorts in the public domain (e.g. TCGA, cBioportal).

Response: Relevant data are not available in public database, like TCGA, cBioportal etc.

Comment: There is no extensive bio-informatic or (even preliminary) functional analysis on the gene variants found. No discussion is provided for the variants from the literature (also from other cancers?).

Response: We inserted an additional column in the Table 2. It contains the pathogenicity scores obtained by fitCons tool; it is in silico pathogenicity predictor, which integrates the results of relevant functional assays (such as ChIP-Seq) with the data on selective pressure. We also provide some overview on the PRF1 p.Ala91Val in the Discussion:

The hypomorphic p.Ala91Val allele (rs35947132) is the most common variant observed in the Caucasian population, with a minor allele frequency (MAF) of 7–9%. The pathogenic potential of this mutation is well documented. The monoallelic p.Ala91Val substitution results in a 10-fold reduction in the lytic activity of the enzyme due to protein misfolding, leading to a notable decline in NK-cell cytotoxicity in healthy carriers of this variant [34,35]. Individuals who are homozygous for p.Ala91Val develop familial hemophagocytic lymphohistiocytosis type 2 (FHL2), although the penetrance of this genotype is not complete [36]. Some data indicate that PRF1 p.Ala91Val heterozygosity is associated with subclinical immunodeficiency symptoms [37]. Notably, PRF1 p.Ala91Val has also been linked to an increased risk of various hematological cancers, including B and T-cell lymphoma and acute lymphoblastic leukemia [38-40].

Comment: This makes it difficult to re-construct how these variants could influence the tumorigenesis.

Response: We now provide some relevant speculations in the Discussion:

“One may hypothesize that in young females, where hormonal stimulation drives rapid mammary epithelial proliferation, the combined effects of defective DNA repair resulting from BRCA1 dysfunction and compromised immune surveillance due to PRF1 deficiency may synergistically accelerate oncogenic transformation. Furthermore, subclinical impairment of natural killer cell function and other cytotoxic effector mechanisms associated with PRF1 p.Ala91Val heterozygosity may promote the release of pro-inflammatory cytokines, including TNF, IL-6, and IFN-γ [41,42], fostering a microenvironment conducive to tumor initiation and progression. The elevated incidence of hematologic malignancies among PRF1 p.Ala91Val carriers further supports the broader oncogenic potential of partial perforin deficiency [38–40].”

Comment: The Materials and Methods section is way too short & superficial.

Response: We have extended this section.

Comment: The figures, though descriptive, are large and not consistent in style, font, etc. This makes comparison difficult.

Response: We have revised Figures according to this suggestion.

Round 2

Reviewer 2 Report

Comments and Suggestions for Authors

The authors have addressed my concerns in a satisfactory manner. The manuscript is now suitable for publication.